# Maximizing multi-reaction dependencies provides more accurate and precise predictions of intracellular fluxes than the principle of parsimony

**Seirana Hashemi**[1], **Zahra Razaghi-Moghadam**[1,2], **Zoran Nikoloski**[1,2]*

**1** Bioinformatics Department, Institute of Biochemistry and Biology, University of Potsdam, Potsdam, Germany, **2** Systems Biology and Mathematical Modeling Group, Max Planck Institute of Molecular Plant Physiology, Potsdam, Germany

* nikoloski@mpimp-golm.mpg.de

**Data Availability Statement:** All data and code used for running the simulations and analyses is available on a GitHub repository at https://github.com/seirana/cbFBA and is based on MATLAB.

## Abstract

Intracellular fluxes represent a joint outcome of cellular transcription and translation and reflect the availability and usage of nutrients from the environment. While approaches from the constraint-based metabolic framework can accurately predict cellular phenotypes, such as growth and exchange rates with the environment, accurate prediction of intracellular fluxes remains a pressing problem. Parsimonious flux balance analysis (pFBA) has become an approach of choice to predict intracellular fluxes by employing the principle of efficient usage of protein resources. Nevertheless, comparative analyses of intracellular flux predictions from pFBA against fluxes estimated from labeling experiments remain scarce. Here, we posited that steady-state flux distributions derived from the principle of maximizing multi-reaction dependencies are of improved accuracy and precision than those resulting from pFBA. To this end, we designed a constraint-based approach, termed complex-balanced FBA (cbFBA), to predict steady-state flux distributions that support the given specific growth rate and exchange fluxes. We showed that the steady-state flux distributions resulting from cbFBA in comparison to pFBA show better agreement with experimentally measured fluxes from 17 *Escherichia coli* strains and are more precise, due to the smaller space of alternative solutions. We also showed that the same principle holds in eukaryotes by comparing the predictions of pFBA and cbFBA against experimentally derived steady-state flux distributions from 26 knock-out mutants of *Saccharomyces cerevisiae*. Furthermore, our results showed that intracellular fluxes predicted by cbFBA provide better support for the principle of minimizing metabolic adjustment between mutants and wild types. Together, our findings point that other principles that consider the dynamics and coordination of steady states may govern the distribution of intracellular fluxes.

**Funding:** DFG project NI 1472/4-2 to Z.N. European Union's Horizon 2020 research and innovation programme grant 862201 to Z.N. The funders had no role in study design, data collection and analysis, decision to publish, or preparation of the manuscript.

## Author summary

Data on intracellular fluxes in biological systems provide a snapshot of the rates of underlying reactions and activity of metabolic pathways. However, capturing the activity of reactions and pathways is very resource-intensive, precluding widespread usage of fluxes in understanding of cellular physiology. Therefore, approaches for accurate and precise prediction of intracellular fluxes can propel the usage of intracellular fluxes in diverse biotechnological application that require the identification of reaction targets. Here, we propose a constraint-based approach, termed complex-balanced flux balance analysis, based on the principle of maximizing multi-reaction dependencies. By using data sets of intracellular fluxes in strains of two model organisms, *Escherichia coli* and *Saccharomyces cerevisiae*, we show that the predictions from our approach are more accurate and precise in comparison to a widely used approach relying on the principle of parsimonious usage of cellular resources. Therefore, our results suggest that other cellular principles, related to properties of steady state fluxes, such as multi-reaction dependencies, may shape cellular physiology.

## Introduction

Advances in constraint-based metabolic modeling have resulted in increased accuracy of predictions for diverse molecular phenotypes [1] and cellular functions [1]. The predictions of intracellular fluxes and other metabolic phenotypes in the constraint-based modeling framework depend on the objective function employed in narrowing down the space of steady-state flux distributions. For instance, flux balance analysis (FBA) [2]—the first representative of this modeling framework—relies on optimizing the biomass objective function that models the specific growth rate via a synthetic biomass reaction [3]. FBA has been instrumental for the accurate prediction of specific growth rates and exchange fluxes not only in prokaryotes, like the model bacterium *Escherichia coli* [4], but also in eukaryotes, like *Saccharomyces cerevisiae* [5,6], *Arabidopsis thaliana* [7], and *Homo sapiens* [8]. However, FBA and its successors require additional input to boost the performance in predicting intracellular fluxes, both with respect to accuracy and precision, under different environments, for different genotypes, and their combinations [9–11].

To this end, three directions have been followed to improve predictions of intracellular fluxes by constraint-based modeling: (i) developing approaches for automated identification of objectives that a cellular system optimizes by using data for experimentally estimated fluxes obtained by integrating metabolomics data from labeling experiments with (smaller) metabolic models [12,13]; these approaches are cast as bilevel programming problems by identifying a linear combination of reaction fluxes already included in the metabolic network or of fluxes for newly introduced demand reactions, (ii) investigating the trade-off between well-studied objectives in a multi-objective setting [14] or comparing multiple objectives [15]; for instance, following this approach, it has been shown that intracellular fluxes in *Escherichia coli* may arise due to the optimization of growth, maximization of ATP production, and minimization of total flux [14]; this is in line with an earlier study showing that there seems not to be a single objective the optimization of which leads to good predictions under different scenarios (e.g. nutrient-rich vs. nutrient-poor conditions [15]) and (iii) performing iterative optimization with several objective functions over progressively restricted feasibility regions; for instance, optimization of biomass objective function, followed by minimization of total flux at a fraction of the optimal specific growth—typical for the approach called parsimonious (enzyme usage)

FBA (pFBA) [16]. These approaches, catalogued and categorized in [17], are part of the constraint-based modeling framework based on FBA.

pFBA is based on the principle that a cell minimizes the total necessary enzyme mass to implement an optimal specific growth rate. It has been implemented in two variants, where the minimization is for the total flux of gene-associated reactions [16] or all reactions in the network. Since the application of pFBA usually results in a substantial narrowing of the solution space, as indicated by a variant of flux variability analysis [18], it has been widely used in different applications. For instance, it has been employed to arrive at a single flux distribution for specific growth scenarios in model organisms, which are subsequently compared [19–21], or is used in the estimation of enzyme parameters [18,22,23], where the specific growth rate along with other exchange fluxes with the environment are constrained based on experimental measurements. However, there has been no quantitative comparison between intracellular fluxes obtained by pFBA with experimentally estimated fluxes from labeling experiments. Most studies that rely on pFBA and consider flux estimates from labeling experiments usually involve flux fitting [24] or constraints from the confidence intervals that are inherent to the flux estimates. An additional concern is that the flux estimates from labeling experiments are usually obtained by using smaller metabolic models than those considered in constrained-based modeling with pFBA, shown to affect the flux estimates [25]. As a result, the predictions from these variants of pFBA may bias the comparison of the resulting predictive performance.

A direction that has not yet been considered in the prediction of steady-state flux distributions deals with the dynamical properties of the respective steady states. For instance, chemical reaction network theory [26] has been able to associate properties of the underlying network structure with particular features of the dynamics supported by the network. This has led to the development of approaches for model reduction [27] and the investigation of multi-reaction dependencies in metabolic networks [28]. Here, we investigate if principles related to maximizing multi-reaction dependencies result in improved accuracy and specificity of intracellular flux predictions in comparison to pFBA in model organisms.

## Results and discussion

### Complex-balanced FBA incorporates the principle of maximizing multi-reaction dependencies

To explain the formulation of the proposed approach, termed complex-balanced FBA (cbFBA), we first introduce the concepts used in constraint-based metabolic modeling. A metabolic network represents a set of biochemical reactions (**Fig 1A**) through which nutrients obtained from the environment are transformed into metabolic intermediates (i.e., metabolites) and then into the building blocks of biomass. Note that in an open system, some metabolites are exchanged with the environment, here denoted by O. For instance, reaction $E_1$ on **Fig 1A** takes up metabolite A. Every biochemical reaction has a left- and right-hand side that represent non-negative linear combination of the metabolites in the network. The coefficients of these linear combinations correspond to the stoichiometry with which metabolites participate in reactions. The left- and right-hand sides of every reaction are considered as complexes, a concept arising from chemical reaction network theory [26]. Thereby, every reaction can be represented as a directed edges that connected two nodes corresponding to the substrate and product complexes, respectively. For instance, reaction $R_1$ on **Fig 1A** has A as its substrate complex and B+C as its product complex.

The structure of a metabolic network can then be described by a directed graph in which the nodes correspond to the complexes, denoting the left- and right-hand sides of the reactions, and directed edges correspond to irreversible reactions with their corresponding

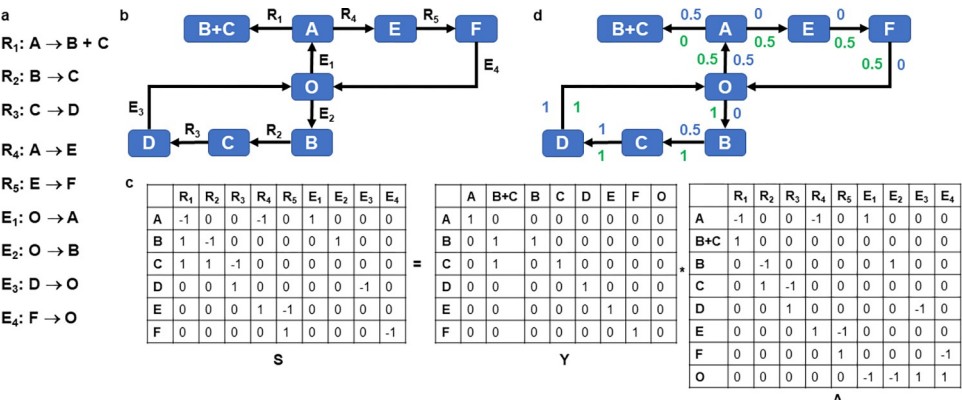

**Fig 1. A metabolic network with six metabolites and nine irreversible reactions. a**. Set of nine irreversible reactions that transform six metabolites, named A—F; O refers to the environment, in this example borrowed from [31]. For of the reactions are exchange reactions, denoted by $E_1$ to $E_4$, while the remaining reactions are named $R_1$ to $R_5$; all reactions are depicted by arrows. **b**. The complexes of the model with incoming and outgoing reactions. **c**. matrices, **S**, **Y**, and **A**. **d**. Two different steady states with different balanced complexes, resulting from cbFBA (in green) and from pFBA (in blue), along with the corresponding flux values.

substrate and product complexes. For instance, for the network of eight reactions in **Fig 1A**, there are eight complexes (**Fig 1B**). As a result, the stoichiometric matrix, **S**, that gathers the stoichiometric coefficients with which metabolites enter as substrates (negative) and products (positive), can be decomposed into the matrix **Y**, capturing the stoichiometry with which species enter complexes, and the incidence matrix **A** if the directed graph (**Fig 1C**), i.e., **S** = **YA**. From the explanation above, given a set of biochemical reaction it is easy to obtain the incidence matrix, **A**, of the directed graph representation. The complexes of every reaction can also be readily obtained by considering the sign of the reaction vector in the stoichiometric matrix. For instance, reaction $R_1$ on **Fig 1C** has one -1 entry for metabolite A, yielding the complex A; it has two entries of 1 for metabolites B and C, yielding the complex B+C (see Methods).

Metabolic networks are usually studied at steady state, whereby the concentrations of internal metabolites, **x**, do not change with time, i.e. $\frac{d\mathbf{x}}{dt} = \mathbf{S}\mathbf{v} = \mathbf{Y}\mathbf{A}\mathbf{v} = \mathbf{0}$. The outcome of these analyses comprises the steady-state reaction fluxes gathered in the steady-state flux distribution, **v**. The activity of a complex $i$ is then given the difference between the total flux of reactions that have complex $i$ as a product and the total flux of reactions that use complex $i$ as a substrate, i.e. $\mathbf{A}_{i.} \mathbf{v}$, where $\mathbf{A}_{i.}$ denotes the i-th row of the matrix **A** [27,28]. Based on the concept of activity, we can distinguish balanced from non-balanced complexes. A complex $i$ is called balanced if its activity is zero for any steady-state flux distribution, **v**, then $\mathbf{A}_{i.} \mathbf{v} = 0$; otherwise, it is considered non-balanced. As a result, the sums of reaction fluxes using the complex $i$ as a substrate and product, respectively, are the same. For instance, complex D on **Fig 1B** is balanced for any steady state flux distribution the network obtains, since metabolite D appears in only this complex (i.e., the activity of this complex coincides with the steady-state constraint for concentration of metabolite D). In addition, the complex B+C is non-balanced for any positive steady-state flux distribution the network obtains, since this complexes only has incoming reactions. Linear programming can be used to efficiently determine such balanced complexes (**Materials and Methods**) [27,28]. Since for a balanced complex, the sum of fluxes of the incoming reactions is equal to the sum of fluxes of outgoing reactions, balanced complexes imply multi-reaction dependencies that cannot be readily read out from the steady-state equations and arise due to interplay of metabolites in complexes and reactions of the network.

Here we expand on this notion of balanced complex, and define a complex balanced for the steady-state flux distribution, **v** if the activity of the complex is balanced for that **v.** This notion also implies dependencies between the multiple incoming and outgoing reactions from the complex $i$, but for the steady-state flux distribution **v** rather than the entire flux space specified by the constraints imposed. For instance, balanced complexes differ between the two steady-state flux distributions (**Fig 1D**), one shown in blue and the other depicted in green. The balanced complexes for first steady-state flux distribution pFBA include the complexes A, E, F, D, while the balanced complexes for the second flux distribution include A, B, C, D, E, F, and O. Note that the complexes A, E, F, D are balanced over all steady-state flux distributions. Moreover, per definition, the second flux distribution implies more multi-reaction dependencies, as specified by the activities of the complexes balanced for this flux distribution.

We postulate that upstream transcriptional and transcriptional regulation of real-world metabolic networks leads to the maximization of multi-reaction dependencies as a principle of metabolic functionality; these multi-reaction dependencies would decrease the need for fine-tuned regulation of reaction fluxes. Multi-reaction dependencies are also linked to the modular structure of metabolic and other biological networks [29], providing further plausibility of this principle. This idea clearly differs from the principle of metabolic functionality rooted in enzyme costs, like pFBA. For instance, the flux distribution depicted in blue on **Fig 1D** results from pFBA, while that shown in green is obtained by maximizing the multi-reaction dependencies, as visible in the larger number of balanced complexes for the latter. This toy network illustrates the notable differences between pFBA and cbFBA.

To mathematically model the principle of maximizing the number of multi-reaction dependencies, we propose that steady-state flux distributions are determined by maximizing the number of balanced complexes in a given flux distribution, which we approximate by solving a linear programming problem (**Materials and Methods**). Note that exact maximization of the number of balanced complexes is infeasible since this corresponds to the minimization of the zero norm of **A** [30]. The constraint-based formulation facilitates the integration of uptake, secretion, and specific growth rates allowing us to test the extent to which predictions from cbFBA coincide with estimates from $^{13}$C-MFA.

## cbFBA improves the precision of predicted flux distributions for *E. coli* strains compared to pFBA

To verify the proposed principle of metabolic functionality based on multi-reaction dependencies, we applied cbFBA and pFBA with constraints from uptake, secretion, and specific growth rates of 17 *E. coli* evolutionary adapted knock-out strains [32–35] to predict their steady-state flux distributions. Single steady-state flux distributions resulting from solving the linear programming problems (**Materials and Methods**) do not provide insights into the space of alternative solutions. To resolve this issue, we determined flux ranges from cbFBA and pFBA [18] by extending flux variability analysis [36] (**Materials and Methods**, **S1 Table**). Further, to obtain robust estimates of the mean fluxes in the respective flux ranges, we generated 1000 steady-state flux distributions under the constraints of cbFBA and pFBA by flux sampling (**Materials and Methods**). These predictions and computations allowed us to make use of the available means and confidence intervals for intracellular fluxes for 565 reactions available from $^{13}$C-MFA [32–35] (**S2 Table**) in our comparative analyses.

The comparison of predicted flux ranges and mean fluxes against the experimental confidence intervals and mean values can be conducted in a qualitative and quantitative fashion. The qualitative comparison entails determining the agreement between the activity patterns of reactions and correlations between the fluxes of reactions predicated and measured as active;

further, the correlations consider log-transformed flux values to account for the orders of magnitude differences in flux values in steady-state flux distributions. In contrast, the quantitative agreement of flux values is carried out by considering the agreement between the means and the overlap of flux ranges (in the space of alternative solutions).

First, we determined the concordance of the reaction status—blocked or active—for those reactions for which fluxes were available from both predictions and experiments. To this end, to avoid numerical issues, we considered a reaction to be blocked if the flux was below $10^{-6}$ mmol gDW$^{-1}$ h$^{-1}$, as performed in other studies [37–39]. We did not observe a difference in the average sensitivity (0.99) nor the average specificity (0.46) between cbFBA and pFBA, resulting in the same average accuracy of 0.97 for the predicted active reactions (**S3 Table**).

Furthermore, with respect to the flux variability analysis (**Materials and Methods**), we found that both pFBA and cbFBA do not result in unique flux distributions, due to the presence of variable fluxes at the respective optima for the two approaches. Further, we found that the flux ranges from experiments overlapped with the flux ranges from cbFBA and pFBA on average for 77% and 93% of reactions with measured fluxes (**S3 Table** and **Fig 2**). This finding may suggest that pFBA is a better alternative to the proposed cbFBA; however, this qualitative comparison does not suffice to draw such a conclusion. Interestingly, the reactions with

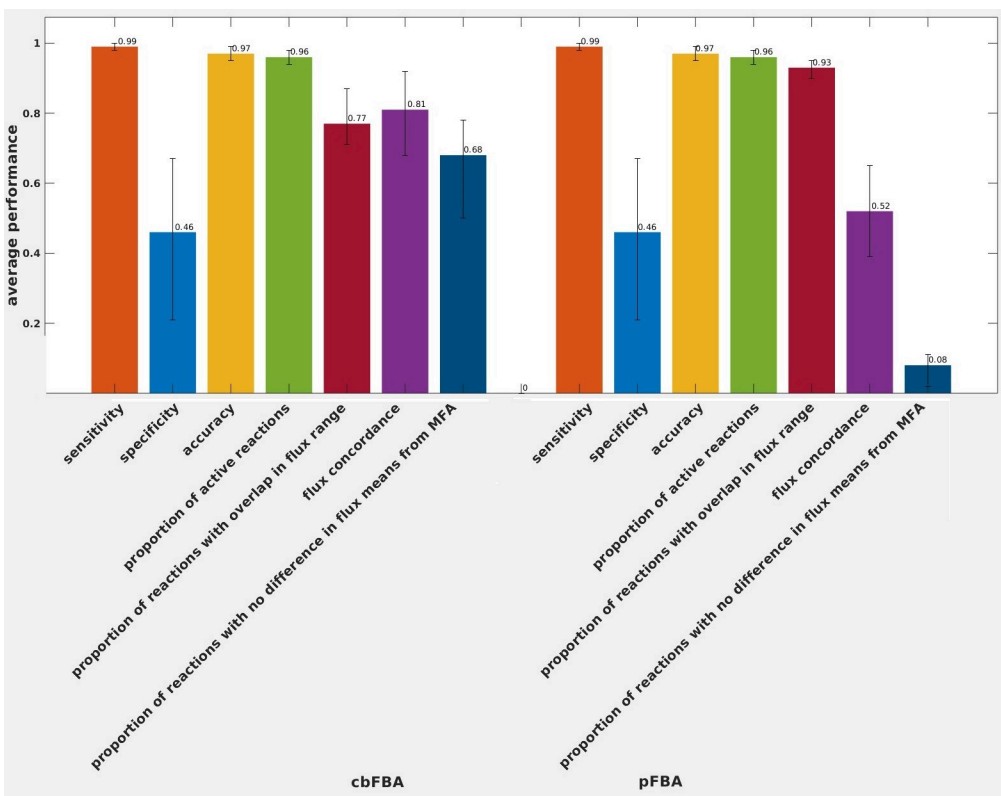

**Fig 2. Comparison of performance of cbFBA and pFBA in the case of *E. coli*.** The first three bars show the sensitivity, specificity, and accuracy of predicted active reactions in comparison to the experimentally determined active reactions. The fourth bar shows the fraction of active reactions, while the fifth bar illustrate the fraction of reactions whose predicted flux ranges overlap with confidence intervals from experiments. Flux concordance refers to the Pearson correlation between log-transformed fluxes of reactions deemed active in pFBA, cbFBA, and experimental flux distributions; the last bar refers to the fraction of reactions with flux estimates that showed no difference to the means determined by cbFBA and pFBA, respectively. The error bars denote the minimum and maximum values that these measures of performance over the 17 *E. coli* strains.

predicted flux ranges that overlapped with those from experiments differed between the two approaches. Notably, reactions that showed overlaps between flux ranges from experiments and predictions exclusive to cbFBA were enriched in Alanine and Aspartate Metabolism, Alternate Carbon Metabolism, Citric Acid Cycle, Extracellular exchange, Glycine and Serine Metabolism, Histidine Metabolism, Inorganic Ion Transport and Metabolism, Nucleotide Salvage Pathway, Pentose Phosphate Pathway, Purine and Pyrimidine Biosynthesis, Pyruvate Metabolism, and Transport, Outer Membrane Porin. These findings indicate that cbFBA predictions for rates in respiratory metabolism and amino acid metabolism along with uptake and excretion are in line with evidence from [13]C-MFA. In addition, reactions that showed overlapping ranges between experiments and predictions only from pFBA were enriched in Arginine and Proline Metabolism, Cell Envelope Biosynthesis, Cofactor and Prosthetic Group Biosynthesis, Extracellular exchange, Glycerophospholipid Metabolism, Glycolysis / Gluconeogenesis, Lipopolysaccharide Biosynthesis / Recycling, Membrane Lipid Metabolism, Murein Biosynthesis, Nucleotide Salvage Pathway, Pentose Phosphate Pathway, Transport, Inner Membrane, and Transport, Outer Membrane Porin. These findings suggest that pFBA predictions are in line with ranges from [13]C-MFA for the remaining pathways involved in respiration.

To further illustrate the comparison between these findings and stress the precision of the flux predictions from cbFBA, we grouped the reactions based on how the flux ranges predicted from the variability analysis from cbFBA compare to the flux ranges predicted from the variability analysis from pFBA (**Fig 3A**, **S4** and **S5 Tables**). We found that 73.75% of reactions showed higher maximum flux in pFBA in comparison from that in cbFBA, while maintaining the same minimum flux. In addition, 16.96% of reactions demonstrated larger flux range in pFBA in comparison to cbFBA. Small percentage of reactions (8.5%) showed higher maximum and minimum flux from pFBA in comparison to cbFBA, with or without the overlap in the ranges. Lastly, only 0.73% of reactions showed the same flux range in pFBA as in cbFBA. Interestingly, the logarithm of the difference between the maximum and minimum attainable fluxes resulting from cbFBA was consistently negative for reactions across all metabolic subsystems

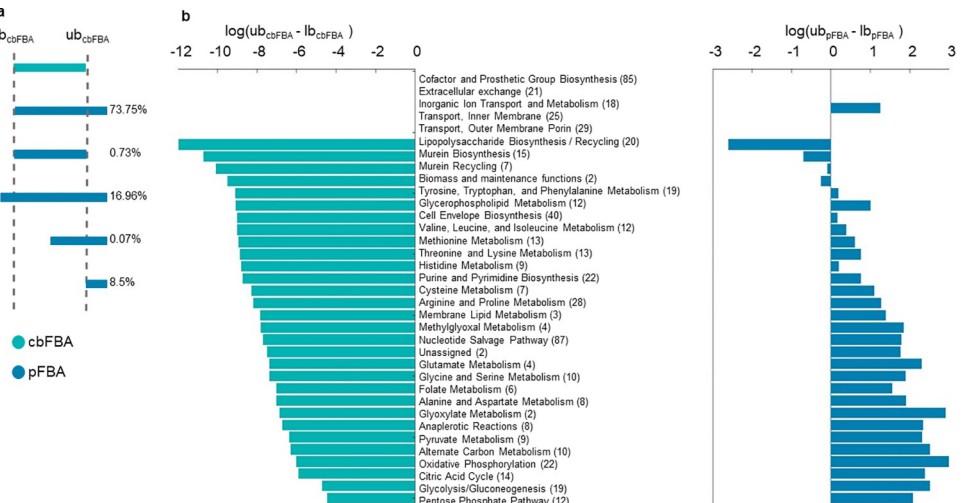

**Fig 3. Categorization of reactions based on flux ranges from pFBA in comparison to cbFBA. a.** The reactions can be categorized based on how the flux range from pFBA (blue) compares to the flux range from cbFBA (in green). **b.** Flux range, given by the logarithm of the maximum and minimum fluxes, from cbFBA and pFBA are categorized per metabolic subsystems in the model of *E. coli*. Shown are the average flux ranges per metabolic subsystem.

(**Fig 3B**). In comparison, this was only the case for the Lypopolysaccharide Biosynthesis / Recycling, Murein Biosynthesis, Murein Recycling, and, expectedly, Biomass and maintenance functions. These findings demonstrate that the predictions from cbFBA are more precise than those from pFBA since cbFBA results in a smaller space of alternative solutions than pFBA.

### cbFBA improves the accuracy of predicted flux distributions for *E. coli* strains compared to pFBA

To obtain better insights into the concordance of fluxes, we determined the Pearson correlation coefficient for the log-transformed mean flux values of reactions deemed active by cbFBA, pFBA, and experiments. We found that the average Pearson correlation over the 17 strains for cbFBA (0.81) was larger than that of pFBA (0.52) (**S3 Table, Fig 2**). Since flux distributions of high correlation can differ in terms of the magnitude of fluxes, to further investigate the concordance between mean predicted and experimental fluxes, we determined the percentage of reactions with flux estimates that showed no difference to the means determined by flux sampling. We found that for an average of 67% and 8% of these reactions, cbFBA and pFBA, respectively, did not show statistically significant differences in mean fluxes over the considered strains (Spearman, mFDR correction, **S3 Table**). In the case of active reactions over the three approaches, we found that cbFBA and pFBA showed no difference, on average, for 68% and 8% of reactions (**Fig 2**). These findings demonstrate that cbFBA provides better agreement with experimentally measured fluxes in comparison to pFBA for the compared experiments in *E. coli*.

### Categorization of complexes for the different flux distributions in *E. coli*

We also employed the confidence intervals for fluxes obtained from experiments as constraints and determined the balanced complexes for all steady-state flux distributions in this feasible space (**Materials and Methods**). We used the identified balanced complexes to determine the accuracy of cbFBA and compared it to pFBA. The sensitivity and specificity of predicted balanced complexes by cbFBA were, on average, 1 and 0.96, while pFBA resulted in average sensitivity and specificity of 1 and 0.94 over the 17 strains. Therefore, the accuracy of predictions of balanced complexes was quite high and comparable at 0.97 and 0.98 for cbFBA and pFBA, respectively (**S6 Table**).

Complexes with only incoming or outgoing reactions are never balanced in any positive steady-state flux distribution. In addition, complexes that are balanced in the entire flux cone are also balanced in any subset of steady-state flux distributions (**Materials and Methods**). Finally, some complexes are balanced in particular flux distributions (see **Fig 1D**). In our analysis, the difference in the set of balanced complexes for different strains are due to these types of complexes that are balanced for some, but not all flux distributions in the steady-state flux cone. Interestingly, we found that there are only a few (i.e., 77 of 4290) complexes that are balanced in only some of the flux distributions predicted by pFBA and cbFBA. For instance, 32 of such complexes are unbalanced for strains *tp2* and *tp3*, but are balanced in the flux distributions of the other strains. Altogether, the reactions that are incident on the 77 balanced complexes belong to seven different pathways, including: murein recycling, alternate carbon metabolism, cofactor, and prosthetic group biosynthesis, extracellular exchange, inorganic ion transport and metabolism, intracellular demand and transport, inner membrane (**S7 Table**).

Interestingly, complexes that are predicted to be balanced in only some of the steady-state flux distributions by both pFBA and cbFBA (while applying the flux bounds from the confidence intervals) are also balanced in the experimental flux distributions–further supporting the quality of the predictions from the proposed approach.

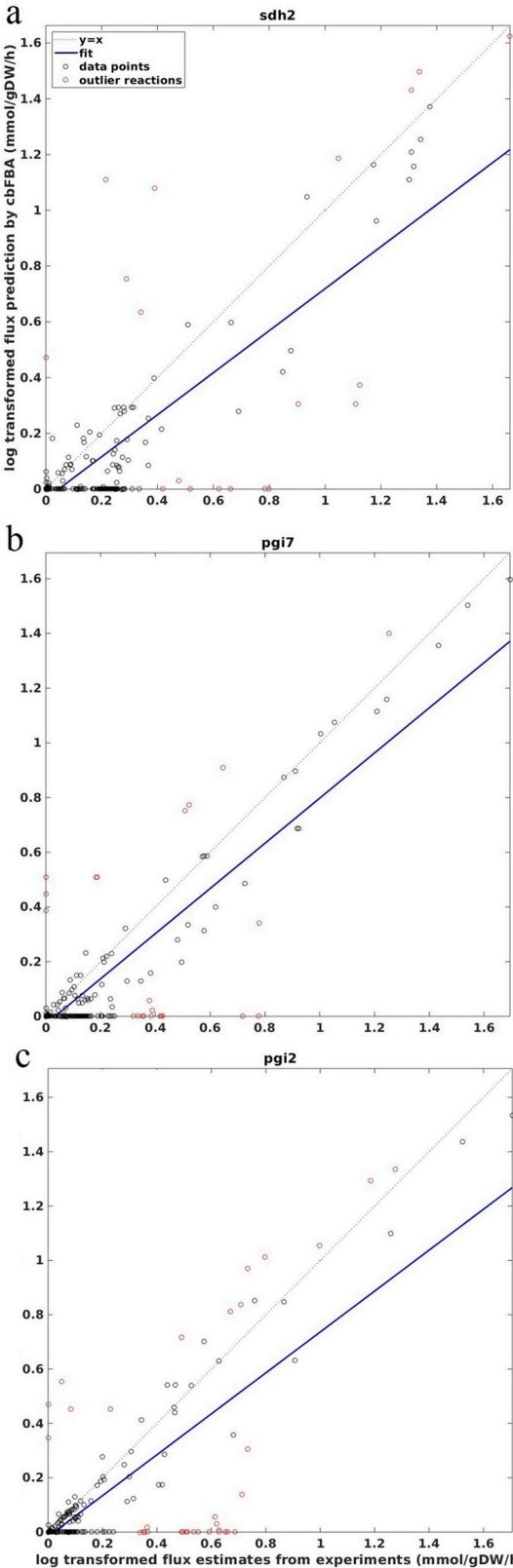

**Fig 4. Outlier reactions for cbFBA.** Using the reactions with both experimentally determined and predicted values, we plotted the strains with (a) the smallest, (b) median, and (c) largest number of outliers. Outliers are denoted by red dots in the log-log plot of predicted and experimentally determined fluxes. Dashed line indicates correlation of one.

## Outlier reactions based on predicted and estimated fluxes

The presented results about correlation between log-transformed predicted and estimated fluxes are affected by outlier reactions. These are the reactions for which the predicted and estimated fluxes drastically disagree. To characterize these outliers, we determined the reactions whose error of prediction (i.e., distance from the regression line) is twice the average sum of squares error (i.e., $s = \sqrt{\sum (\hat{y} - y)^2 / (n - 2)}$). The strains with the smallest number of outlier reactions include sdh2 (and sdh3) (**Fig 4A**), while those with the largest number of outliers was pgi2 (**Fig 4C**); strains like *pgi7* and *pgi1* show median / mean number of outliers (**Fig 4B**). We found that the phosphoenolpyruvate carboxykinase and oxaloacetate transport were identified as outliers across all strains (**S18 Table**). Note that these analyses were conducted by considering log-transformed flux values, to account for the order of magnitude differences between flux values in a steady-state flux distribution.

## cbFBA improves the predictions of flux distributions for *S. cerevisiae* strains

We applied cbFBA and pFBA with uptake constraint for D-glucose and secretion for ethanol, glycerol, acetate, pyruvate, and succinate, and specific growth rates of 26 *S. cerevisiae* mutants [5] (**S8 Table**) to predict steady-state flux distributions. We determined flux ranges from cbFBA and pFBA (**S9 Table**). Further, to obtain robust estimates of the mean fluxes in the respective flux ranges, as in the analysis of *E. coli* strains, above, we generated 1000 steady-state flux distributions under the constraints of cbFBA and pFBA by flux sampling (**Materials and Methods**).

These computations allowed us to conduct a comparative analysis with the available means and confidence intervals for intracellular fluxes for 27 intracellular reactions (**S10 Table**). Of these reactions that are active in all mutants, one reaction, namely nitric oxide dioxygenase, was predicted as blocked by both methods, cbFBA, and pFBA, in the flux distributions of all mutants. Therefore, the sensitivity, specificity, and accuracy of predicting active reactions for pFBA and cbFBA were the same (**S9 Table**).

Concerning the flux variability analysis, we found that the flux ranges from experiments overlapped with the flux ranges from cbFBA and pFBA on average for 53% and 39% of reactions with measured fluxes (**S11 Table**). Interestingly, for the reactions identified as active in experiments, we observed that on average, seven of reactions showed smaller minimum flux in pFBA in comparison to cbFBA, while on average, 76% of reactions showed larger maximum flux in pFBA in comparison to cbFBA (**S12 Table**). Finally, we found that the ranges predicted by cbFBA were fully contained in the flux ranges predicted by pFBA for 72 reactions, on average 72% (**S13 Table**). Similar to the case of *E. coli*, this finding demonstrates that the predictions from cbFBA are more precise than those from pFBA since the former approach has a smaller alternative space than the latter.

To follow up the qualitative analyses, we next determined the Pearson correlation coefficient for the log-transformed mean flux values of reactions deemed active by cbFBA, pFBA, and experiments. We found that the average Pearson correlation over the 26 mutants for cbFBA was 0.94, slightly larger than that of pFBA (0.90) (**S11 Table**). Finally, we determined the percentage of reactions with flux estimates that showed no difference from the means determined by flux sampling. We found that ~ 52% of active reactions did not show statistically significant differences in mean fluxes over the considered mutants by both approaches (**S11 Table**).

Finally, we employed the confidence intervals for fluxes obtained from experiments as constraints and determined the balanced complexes for all steady-state flux distributions in this

space. Then, we used the identified balanced complexes to determine the accuracy of cbFBA and compare it to pFBA. Our findings showed that cbFBA consistently has more balanced complexes than pFBA. Sensitivity and specificity for predicting balanced complexes of 1 and 0.98 over the 26 strains, while pFBA resulted in average sensitivity and specificity of 0.99 and 1, respectively. Therefore, the accuracy of predictions of balanced complexes was the same for cbFBA and pFBA in the case of *S. cerevisiae* strains (**S14 Table**).

## Flux distributions of cbFBA are in better agreement with the principle of minimization of metabolic adjustment

To provide further evidence for the improved prediction of intracellular fluxes from cbFBA, we determined the Euclidean distance between the flux distributions of the mutants and the wild type resulting from either cbFBA or pFBA. To this end, we sampled 1000 pairs of steady-state flux distributions compatible with confidence intervals from ${}^{13}$C-MFA, on one hand, and the constraints from pFBA and cbFBA, on the other hand. This allowed us to obtain a distribution of the Euclidean distances for each strain from cbFBA and pFBA as well as the distribution of the robust mean Euclidean differences (**S15 Table**). We found that the Euclidean distance between the flux distributions from cbFBA was on average smaller than that resulting from pFBA for both *E. coli* and *S. cerevisiae*. Therefore, the intracellular fluxes predicted from cbFBA better support the principle of minimizing metabolic adjustment [40] in mutants in comparison to pFBA.

## Conclusion

Despite advances in the constraint-based modeling framework, the accurate prediction of intracellular fluxes remains an important problem. Addressing this problem is very relevant, as it provides easier means to quantify intracellular fluxes under specific environments, which can in turn, be used for parameterization of enzyme kinetics [41,42]. pFBA has now been widely used to decrease the space of steady-state flux distribution that support particular specific growth rate and exchange fluxes. While intracellular flux estimates from labeling experiments using ${}^{13}$C-MFA [43] provide the means to assess the accuracy and precision of pFBA predictions, these estimates are usually used as constraints to constrain the steady-state flux distributions.

Here, we argued that principles related to the coordination of flux values, reflected in multi-reaction dependencies, may provide another, yet unused, means to predict intracellular flux distributions. The concept of balancing of complexes imposes equalities between the sums of the flux of incoming and outgoing reactions either in the entire flux cone [27] or particular steady-state flux distributions, depending on the restrictions imposed. Balancing of complexes in a given flux distribution reflects coordination on flux distributions imposed by upstream transcription- and translation-related cellular processes. We showed that maximization of such multi-reaction dependencies due to balancing could be captured in a linear programming problem, which we term complex-balanced FBA, cbFBA. In contrast to previous studies that have focused on evaluating the predictions from different objectives for a limited number of fluxes (e.g., for ten reactions [15]), we made use of recently obtained flux distributions for a considerably larger set of reactions to qualitatively and quantitatively assess the predictions of cbFBA.

Our extensive comparative analyses against pFBA, the most widely used approach for arriving at a representative steady-state flux distribution, with flux distributions from 17 strains of *E. coli* and 26 *S. cerevisiae* knock-out mutants demonstrated that: (i) the accuracy of predicting active/inactive reactions of cbFBA and pFBA are comparable, (ii) cbFBA results in improved precision of predictions in comparison to pFBA, resulting from the smaller set of alternative solutions, (iii) there are more overlaps between confidence intervals from experiments and

intervals of alternative solutions for cbFBA than pFBA, and (iv) the means of predicted flux distributions from cbFBA are in better agreement with estimates from labeling experiments than those from pFBA. In addition, we showed that flux distributions obtained by cbFBA better agree with the principle of minimizing metabolic adjustment that is postulated to underpin flux redistribution between wild-type and mutants. Therefore, we concluded that the principle of maximizing the multi-reaction dependencies, reflected in the balancing of complexes in particular flux distributions, provides the means to predict accurate and precise intracellular fluxes in unicellular model organisms.

Future work in search of principles that can improve the prediction of intracellular steady-state flux distributions can examine additional multi-reaction dependencies [28], opening the possibility for expanding the applicability of techniques from the constrain-based modeling framework beyond the investigation of cellular objectives related to specific growth rate, biomass yield, and exchange fluxes.

## Materials and methods

### Models and data used

Our comparative analysis uses data from two model organisms, *Escherichia coli* strain K-12 substrain MG1655 and *Saccharomyces cerevisiae*, along with their recent large-scale metabolic models. In the case of *E. coli*, we used the iJO1366 model, containing 2583 reactions, 1805 metabolites, and 1365 genes [44]. To perform simulations and compare the findings to flux distributions estimated from $^{13}$C-MFA, we used the published uptake, secretion, and growth rates as well as the estimated fluxes for 17 evolutionary adapted knock-out strains (**S16 and S17 Tables**) [31–34]. For *S.* cerevisiae, we employed the genome-scale metabolic network, yeastGEM v8.3.3, which includes 2691 metabolites and 3963 reactions [45]. To perform simulations and compare the findings with flux distributions estimated from $^{13}$C-MFA, we made use of the glucose uptake rate and selected intracellular fluxes available for 26 knock-out mutants as a function of [5]. The fluxes were estimated using a smaller yeast model, iLL672, with 672 metabolites and 1195 reactions; in our analyses, we employed 26 mutants for which we have data about fluxes of seven extracellular reactions (**S8 Table**).

### Decomposition of a stoichiometric matrix S into YA

```
Input: stoichiometric matrix S∈ℤᵐˣʳ, with m metabolites and r
reactions
Output: metabolite-complex matrix Y∈ℤ₊ᵐˣᶜ, incidence matrix A∈{--
1,0,1}ᶜˣʳ, such that S = YA, where c denotes the number of complexes.
Initialize Y as a matrix with m rows and no columns
R ← array[1..r,1..2] of zeros
for i ← 1 to r do
  h ← array[1..m] of zeros
  t ← array[1..m] of zeros
  for j ← 1 to m do
    if S[i,j]<0 then
      h[j] ← |S[i,j]|
    else if S[i,j]>0 then
      t[j] ← S[i,j]
  end for
  if Y does contains a column equal to h then
    append h as a column to Y
    c ← c+1
    hi ← c
  else
```

```
    hi ← index of the column that equals complex h in Y
  if Y does contains a column equal to t then
    append t as a column to Y
    c ← c+1
    ti ← c
  else
    ti ← index of the column that equals complex t in Y
  R[i,1] ← hi
  R[i,2] ← ti
end for
A ← array[1..c,1..r] of zeros
for i ← 1 to r do
  A[R[i,1],i] ← -1
  A[R[i,2],i] ← -1
end for
return(Y, A)
```

## Formulation of the complex-balanced FBA (cbFBA)

The proposed constraint-based approach, termed complex-balanced FBA (cbFBA), minimizes the total absolute activity of complexes (and, hence, approximates the maximum number of balanced complexes). By doing so, cbFBA models the conservation of multi-reaction dependencies across different flux solutions as a principle of the functionality of metabolic networks. More specifically, cbFBA is formulated by the following linear programming problem:

$$\min_{\mathbf{v}} \sum_{i=1}^{m} |\mathbf{A}_{i.}\mathbf{v}|$$

s.t.

$$\mathbf{Sv} = \mathbf{YAv} = \mathbf{0} \ (LP1)$$

$$\mathbf{v_{min}} \leq \mathbf{v} \leq \mathbf{v_{max}},$$

where $\mathbf{S}$ is the stoichiometric matrix that can be decomposed as the product of the $\mathbf{Y}$ matrix, giving the species composition of $m$ complexes and the incidence matrix, $\mathbf{A}$, of the directed graph with $m$ complexes and nodes and $r$ reactions as directed edges. Here, $\mathbf{A}_{i.}$ denotes the i-th row of the incidence matrix $\mathbf{A}$.

In contrast to this formulation, parsimonious FBA (pFBA) minimizes the total sum through the network and formulates the following linear programming problem:

$$\min_{\mathbf{v}} \sum_{i=1}^{r} |v_i|$$

s.t.

$$\mathbf{Sv} = \mathbf{0} \ (LP2)$$

$$\mathbf{v_{min}} \leq \mathbf{v} \leq \mathbf{v_{max}}.$$

We would like to note that we apply both approaches to networks in which every reversible reaction is split into two irreversible reactions and from which, for fair analysis, dead-end metabolites and blocked reactions for a particular modeling scenario have been removed.

Note that in both cbFBA and pFBA implementations, we use data on specific growth rates and uptake fluxes as constraints.

## Formulation of flux variability analysis

Let $z$ be the solution to the linear programming problem LP1. Flux variability analysis for cbFBA aims to determine the minimum (maximum) of each flux under the constraints that the summation of activities of complexes equals $z$. More specifically, to determine the range for each flux, we solved the following set of linear programs:

$$\min/\max v_i$$

s.t.

$$\mathbf{Sv} = \mathbf{YAv} = \mathbf{0}$$

$$\mathbf{v_{min}} \leq \mathbf{v} \leq \mathbf{v_{max}} \; (\textbf{\textit{LP3}})$$

$$\sum_{i=1}^{c} |\mathbf{A}_{i.}\mathbf{v}| = z.$$

Similarly, for pFBA, let $z'$ denote the solution to the linear programming problem LP2. To determine the range for each flux, we solved the following set of the linear program:

$$\min/\max v_i$$

s.t.

$$\mathbf{Sv} = \mathbf{0}$$

$$\mathbf{v_{min}} \leq \mathbf{v} \leq \mathbf{v_{max}} \; (\textbf{\textit{LP4}})$$

$$\sum_{i=1}^{r} |v_i| = z'.$$

## Formulation of flux sampling

To sample steady-state flux distributions that satisfy the constraints of LP3, we solve the following quadratic programming problem:

$$\min\|\mathbf{v_{rand}} - \mathbf{v}\|_2^2$$

s.t.

$$\mathbf{Sv} = \mathbf{0} \; (\textbf{QP1})$$

$$\sum_{i=1}^{c} |\mathbf{A}_{i.}\mathbf{v}| = z$$

$$\mathbf{v_{min}} \leq \mathbf{v} \leq \mathbf{v_{max}}.$$

The sampling has two steps. First, we generate a random vector of flux values, $\mathbf{v_{rand}}$, in the bounds $\mathbf{v_{min}}$ and $\mathbf{v_{max}}$. Second, we find the vector $\mathbf{v}$ that satisfies steady-state conditions and minimizes the Euclidean distance to $\mathbf{v_{rand}}$.

## Identification of balanced complexes with given flux ranges

The term balanced complex refers to a complex whose activity is zero in any steady state from a specified set of steady states. To identify such balanced complexes, we calculated the minimum and maximum flux around each complex [27] over given flux ranges (from experiments

or arbitrary bounds, as done in FBA). If these values equal zero, then the complex is balanced. More specifically, we solve the following linear programming to find the balanced complexes:

$$\min/\max \mathbf{A}_{i.}\mathbf{v}$$

s.t.

$$\mathbf{Sv} = \mathbf{YAv} = \mathbf{0} \ (LP5)$$

$$\mathbf{v_{min}} \leq \mathbf{v} \leq \mathbf{v_{max}}$$

where $\mathbf{A}_{i.}$ denotes the i-th row of the incidence matrix $\mathbf{A}$.

## Supporting information

**S1 Table. Lower and upper bound of reactions predicted by cbFBA and pFBA (*E. coli*).**
(XLSX)

**S2 Table. The mean and confidence intervals of intracellular of reactions (*E. coli*).**
(XLSX)

**S3 Table. Comparison of performance of cbFBA and pFBA in the case of *E. coli* regarding active reactions.**
(XLSX)

**S4 Table. Flux range comparison in the case of *E. coli*.**
(XLSX)

**S5 Table. The comparison of ranges predicted by cbFBA and pFBA (*E. coli*).**
(XLSX)

**S6 Table. Comparison of performance of cbFBA and pFBA in the case of *E. coli* regarding balanced complexes.**
(XLSX)

**S7 Table. Incoming and outgoing reactions of complexes that are balanced on some strains and unbalanced on the others (*E. coli*).**
(XLSX)

**S8 Table. Uptake, secretion and biomass rate of strains (*S. cerevisiae*).**
(XLSX)

**S9 Table. Predicted flux ranges for reactions with cbFBA and pFBA methods (*S. cerevisiae*).**
(XLSX)

**S10 Table. The mean and confidence intervals of intracellular of reaction (*S. cerevisiae*).**
(XLSX)

**S11 Table. Comparison of performance of cbFBA and pFBA in the case of *S. cerevisiae* regarding active reactions.**
(XLSX)

**S12 Table. Flux range comparison.** The comparison of the size of flux ranges predicted with the methods cbFBA and pFBA (*S. cerevisiae*).
(XLSX)

**S13 Table. The comparison of ranges predicted by cbFBA and pFBA (*S. cerevisiae*).**
(XLSX)

**S14 Table. Comparison of performance of cbFBA and pFBA in the case of *S. cerevisiae* regarding balanced complexes.**
(XLSX)

**S15 Table. Euclidean distance between the flux distributions between the mutants and WT per cbFBA and pFBA.**
(XLSX)

**S16 Table. Experimental lower bound flux rate for reactions of 17 different strains (*E. coli*).**
(XLSX)

**S17 Table. Experimental upper bound flux rate for reactions of 17 different strains (*E. coli*).**
(XLSX)

**S18 Table. Outlier reactions per strain (*E. coli*).**
(XLSX)

## Author Contributions

**Conceptualization:** Zoran Nikoloski.

**Data curation:** Seirana Hashemi, Zahra Razaghi-Moghadam.

**Formal analysis:** Seirana Hashemi, Zahra Razaghi-Moghadam, Zoran Nikoloski.

**Investigation:** Seirana Hashemi, Zahra Razaghi-Moghadam.

**Methodology:** Seirana Hashemi, Zahra Razaghi-Moghadam, Zoran Nikoloski.

**Project administration:** Zoran Nikoloski.

**Software:** Seirana Hashemi.

**Supervision:** Zahra Razaghi-Moghadam, Zoran Nikoloski.

**Validation:** Zoran Nikoloski.

**Visualization:** Seirana Hashemi, Zoran Nikoloski.

**Writing – original draft:** Zoran Nikoloski.

**Writing – review & editing:** Seirana Hashemi, Zahra Razaghi-Moghadam, Zoran Nikoloski.

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
