## [Decision Letter · Decision Letter 0]

29 Apr 2023

Dear Dr. Nikoloski,

Thank you very much for submitting your manuscript "Maximizing multi-reaction dependencies provides more accurate and precise predictions of intracellular fluxes than the principle of parsimony" for consideration at PLOS Computational Biology.

As with all papers reviewed by the journal, your manuscript was reviewed by members of the editorial board and by several independent reviewers. In light of the reviews (below this email), we would like to invite the resubmission of a significantly-revised version that takes into account the reviewers' comments.

Both reviewers' appreciated the topic and method but raise a number of concerns in regards to the benchmarking the method against other phenotype prediction maximization principles.

We cannot make any decision about publication until we have seen the revised manuscript and your response to the reviewers' comments. Your revised manuscript is also likely to be sent to reviewers for further evaluation.

Sincerely,

Costas D. Maranas

Academic Editor

PLOS Computational Biology

Douglas Lauffenburger

Section Editor

PLOS Computational Biology

Reviewer's Responses to Questions

**Comments to the Authors:**

Reviewer #1: The challenge of estimating accurate metabolic flux maps is addressed in this MS (PCOMPBIOL-D-22-01807) with a new constraint-based method. Accurate flux maps are important for industrial biotechnology, especially for engineered strains or those adaptively evolved in the laboratory to become specialist at the desired production tasks. Genetically, these are strains that have mutations under positive selection pressure with strongly causal mutations. This characteristic in in contrast to WT strains that normally have mutations under negative selection pressure resulting more generalist phenotypes, and thus are probably more flexible use of their metabolic networks so that accurate flux maps are not as critical to know.

Genome-scale models can be used to compute optimal growth and production rates based on flux-balance analysis (FBA) using measured uptake rates. FBA calculations do not give a unique solution for the metabolic flux maps, calling for the imposition of additional constraints to confine possible flux maps. pFBA was developed as a proxy for most parsimonious uses of the metabolic proteome by minimizing the length of the flux vector. Albeit ‘fewer’ and more constrained, the optimal pFBA solutions are not unique. pFBA has become popular method as it is easy to implement and seen to generate physiologically meaningful solutions, consistent with its underlying principles.

The authors aim to generate an improvement of and an alternative to pFBA. They use comparisons to experimental fluxomic data from laboratory-evolved strains as a standard for comparison. It needs to be explicitly recognized that there are experimental errors in these flux estimates. Furthermore, it needs to be recognized that most of the fluxes, reported as fluxomics, are computed from relatively few experimental measurements, typically using a reduced stoichiometric model. Although, I believe that the data the authors use in this paper are from studies that used comprehensive metabolic networks to computational estimate the fluxomic state of the cells. Nevertheless, unlike sequence-based datatypes, metabolomics and fluxomes are datatypes ‘etherical’ in nature as they are very hard to estimate accurately in vivo, and modelers should be aware of these limitations.

The core premise of the proposed and developed new methods is to use topological features of the metabolic network through a decomposition of the stoichiometric matrix, as S=YA, to compute flux states that require minimal regulation, driven by ‘multi-reaction dependences (unfortunately this concept is not clearly and explicity described to be accessible to a general reader).’ The reader is introduced to the method wtih:

“We postulate that upstream transcriptional and transcriptional regulation of real-world metabolic networks leads to the maximization of multi-reaction dependencies as a principle of metabolic functionality; these multi-reaction dependencies would decrease the need for fine-tuned regulation of reaction fluxes.”

Based on this postulate, the authors formulate the complex-balanced FBA (cbFBA) method,

which takes a while to adsorb and understand. I strongly recommend that the authors try to make the concept more accessible and perhaps use a nice graphic/visual to explain it. This visual should ideally include the principles of pFBA as a reference and show how the new methods generates deviation from this state that is understandable based on the underlying principle that they base the method on. Without such clear explanation, the readership of the article and user of the new method may be very few.

The Lewis lab at UCSD maintains a collection of FBA methods: http://cobramethods.wikidot.com/. The cbFBA method, if published, will likely become a part of this collection, and COBRA computational toolboxes. This collection was first published in NRM in, I think 2012. The cbFBA method should be placed and categorized within this structure of methods so the prospective users know how it fits in and understand what its pros and cons are.

Fig 1 is not that easy to understand, and neither are the results of the computations. A small number of modelers will probably make their way through this material, but less expert reader may not. The S=YA is an interesting and thought-provoking decomposition of the stoichiometric matrix. Are the cbFBA solutions truly distinct from the space of allowable pFBA solutions? Are they unique?

pFBA is based on a simple and easy to understand principle, minimization of flux-load, with computations that require minimum number of experimentally measured variables and are a simple confinement of unbounded FBA solutions. The comparison of the pFBA and cbFBA solution spaces should be compared. If they overlap, then the methods have something in common. If they are fully disjointed, then their different characteristics should be delineated. Figure 2 is uninspiring and does not do this subject matter justice.

This methods once clearly described, characterized, and distinguished from pFBA represents publishable material and is suitable for PLoS Comp Bio.

Reviewer #2: Summary:

The author proposed a new variation of flux balance analysis by decomposing the stoichiometric matrix and using one of the decomposed matrices to maximize multi-reaction dependencies. Although the work is not a major improvement from the pFBA algorithm, still it looks interesting. There are a few points that need to be clarified. Besides, the authors can use a smaller model (maybe E. coli core model) to clearly demonstrate a case where cbFBA outsmarts pFBA. Overall, the following points can be addressed to improve the manuscript.

Major Critiques:

1. While authors have made significant effort to justify the claims of the proposed cbFBA’s accuracy and precision over that of pFBA, the explicit contributions and novelty of this scheme need to be further articulated by clearly highlighting knowledge improvement useful to the scientific research community, including a soft biological analysis with relevant background knowledge, otherwise, it would be confounding our understanding of cellular metabolism.

My strong critique would be that minimizing the conservation of multi-reaction dependencies across different flux solutions, other than the parsimony principle, may lead to biologically unrealistic or non-intuitive solutions. The parsimony principle is a widely accepted and commonly used approach in metabolic modeling, particularly with enzyme conservation – as projected application, which assumes that the cell prefers to use the minimum number of reactions or enzymes to carry out a given function. This approach is biologically plausible and has been shown to produce physiologically relevant solutions.

However, minimizing the conservation of multi-reaction dependencies across various flux solutions is a less established method. It may be mathematically elegant, but it is unclear if it is biologically pertinent or consistent with our current understanding of metabolic regulation. In addition, it may generate solutions that are challenging to interpret or experimentally validate. Mismatch in flux overlaps between cbFBA and the other groups, pFBA and experimental data (lines 198–205) suggests a missing link clearly – membrane dysfunctions? Nature would not attempt to balance all complexes, except perhaps signaling molecules. Think of phosphorylation by kinases and several other factors, such as energy efficacy, resource availability, environmental conditions, etc., may also play a role. In certain pathways, maintaining a balance of metabolites and complexes may be essential, but imbalances are often necessary for proper regulation and function. In glycolysis, for instance, ATP production is favored over NADH production, resulting in an imbalanced production of these two metabolites. This imbalance is crucial for the overall function of the pathway and is controlled by numerous enzymes and feedback mechanisms. Also, this is hinged on the model’s completeness.

Moreover, with the postulate that “upstream transcriptional and transcriptional regulation of real-world metabolic networks leads to the maximization of multi-reaction dependencies as a principle of metabolic functionality” (lines 134-136), a potential criticism of the postulate that “maximizing multi-reaction dependencies leads to decreased need for fine-tuned regulation of reaction fluxes” (lines 136-137) is that it may not account for the fact that some reactions in a metabolic network may have more important biological functions than others. For example, some reactions may be critical for cell survival or have regulatory roles, and it may be necessary to fine-tune the fluxes of these reactions even if they are not part of a multi-reaction dependency. Additionally, while linear programming can efficiently determine balanced complexes, it may not be able to capture the full complexity of metabolic networks, which can have non-linear and dynamic behaviors. Therefore, it may be important to validate the postulate with additional modeling and experimental studies. Also, the supporting validation work in Yeast showed some inconsistent trends. Authors would want to justify and should consider that supposed experimental flux estimates from the smaller network as used in this study do not necessarily reflect nature. The reverse would be the case where nature often garners more complexities than model networks.

Therefore, I would recommend that the authors carefully justify and evaluate the use of this approach in their study and provide a clear comparison of the results obtained with the parsimony principle. They should also address the potential limitations and challenges of using this approach and discuss the implications of their findings for our understanding of cellular metabolism.

2. Introduction, lines (90-91): The sentence “since the application of pFBA usually results in a substantial narrowing of the solution space as indicated by a variant of flux variability analysis” contradicts the authors' statement, in lines 30-33 of the abstract - “showed that the steady-state flux distributions resulting from cbFBA in comparison to pFBA show better agreement with experimentally measured fluxes from Escherichia coli strains and are more precise, due to the smaller space of alternative solutions”. Authors should clarify this for the benefit of readers. Moreover, lines 213-216 of result section, says otherwise. All these should be reconciled.

3. Methods, lines 457-469: Potential for bias. While flux sampling in this manner may be redundant, I assume the authors included it to demonstrate the accuracy of cbFBA. However, authors must consider the flux distribution of the sampling vector of random fluxes. It is assumed that authors have used normal sampling, this choice is dependent on the means of flux ranges employed and could be a redundant case, as means of flux samples would likely fall within the means of flux variability range. Consideration of an alternative case of uniform sampling, perhaps to ensure a decent coverage of the entire feasible space of solutions for a large number of samples, is also outside the scope of this paper. In addition, uniform sampling may not be optimal for extremely nonlinear or non-convex optimization problems, as it may overlook crucial regions of the feasible space. In addition to addressing the convergence issue, the popular "coordinate-hit-and-run" algorithm also addresses this one.

4. Methods and Results - Statistical tests and outliers, lines 224-236 & 271-279: To clearly justify the claim that cbFBA improves accuracy, authors must provide additional information or data, bearing in mind that both the Pearson and Spearman's tests used are sensitive to outliers, which can considerably affect the strength and direction of the correlation. The possibility of skewed pFBA predictions may also be related to the biased sampling method.

5. Results and discussions, lines 333-342: The MOMA agreements are dependent on the potential bias of flux mean values. A more accurate analysis of agreement would involve the overlap of flux ranges for both methods and experiments.

6. Results and Discussion, Page 7, Lines 123-127: The decomposition of the S matrix into Y and A is confusing. Please provide a supplementary file and detail each step of the decomposition with an illustrated example. Besides, further discussion is needed regarding the geometrical meaning of each decomposed matrix. For instance, the singular value decomposition of a matrix can be interpreted as rotation, scaling, or rotation. A similar notion is expected for the S=YA decomposition as well.

7. Results and Discussion, Page 7, Lines 129-132: A workflow figure for the S=YA can be useful to understand the concept.

8. Results and Discussion, Page 7, Lines 137-142: As pFBA uses the principles of minimizing the total enzyme requirements in a given metabolic network, the biology behind cbFBA is not clear. Please include your comments on that.

9. Results and Discussion, Page 10, Lines 185-190: If a reaction was blocked in the model and had experimental MFA data available, how did the authors reconcile that?

10. Results and Discussion, Page 10, Lines 192-194: If flux sampling analysis is performed, then what is the relevance of flux variability analysis to determine flux ranges? Please provide an explanation.

11. Results and Discussion, Page 10, Lines 200-209: The authors mentioned several pathways relevant to cbFBA and pFBA. However, what these pathways mean in terms of the biology of E. coli is missing. Please provide an adequate explanation of these pathways.

12. Results and Discussion, Page 10, Lines 217-220: It seems flux range overlap is the main selling point of the cbFBA. However, it is still not convincing the need for flux ranges, when sampling data is available. This raises the question of the better efficacy of cbFBA over pFBA. Please add discussion on this matter.

13. Materials and Methods, Page 16, Lines 412-428: Is the flux space equivalent in both the optimization problem? Please comment on that, if possible, provide proof using mathematical induction/contradiction.

14. Materials and Methods, overall: Provide a detailed procedure of S=YA decomposition.

Minor Critiques:

1. Results and Discussion, 7, Lines 115-117: the figure of the metabolic network can be improved significantly. At first look, this does not even look like a metabolic network, but rather a series of disjoint reactions. An improved version of the figure is expected.

2. Results and Discussion, Page 7, Line 123: “There are ten complexes” or eleven?

3. Materials and Methods, overall: Please comment about the simulation platform.

4. Figure 2, Line 210 and line 227: flux concordance average Pearson correlation should be reconciled. 0.81 on figure and 0.82 in text.

**Have the authors made all data and (if applicable) computational code underlying the findings in their manuscript fully available?**

Reviewer #1: Yes

Reviewer #2: Yes

PLOS authors have the option to publish the peer review history of their article (what does this mean?). If published, this will include your full peer review and any attached files.

Reviewer #1: No

Reviewer #2: **Yes: **Rajib Saha
---

## [Decision Letter · Decision Letter 1]

15 Jul 2023

Dear Dr. Nikoloski,

Thank you very much for submitting your manuscript "Maximizing multi-reaction dependencies provides more accurate and precise predictions of intracellular fluxes than the principle of parsimony" for consideration at PLOS Computational Biology. As with all papers reviewed by the journal, your manuscript was reviewed by members of the editorial board and by several independent reviewers. The reviewers appreciated the attention to an important topic. Based on the reviews, we are likely to accept this manuscript for publication, providing that you modify the manuscript according to the review recommendations. In particular please resolve the final clarification point brought up by Reviewer 2.

Sincerely,

Editorial Board

PLOS Computational Biology

Douglas Lauffenburger

%CORR_ED_EDITOR_ROLE%

PLOS Computational Biology

Reviewer's Responses to Questions

**Comments to the Authors:**

Reviewer #2: The authors need to respond to this.

6. Results and Discussion, Page 7, Lines 123-127: The decomposition of the S matrix into Y

and A is confusing. Please provide a supplementary file and detail each step of the

decomposition with an illustrated example. Besides, further discussion is needed regarding

the geometrical meaning of each decomposed matrix. For instance, the singular value

decomposition of a matrix can be interpreted as

**Have the authors made all data and (if applicable) computational code underlying the findings in their manuscript fully available?**

Reviewer #2: None

PLOS authors have the option to publish the peer review history of their article (what does this mean?). If published, this will include your full peer review and any attached files.

Reviewer #2: No

Figure Files:

Data Requirements:

Reproducibility:

References:

---

## [Decision Letter · Decision Letter 2]

4 Sep 2023

Dear Dr. Nikoloski,

We are pleased to inform you that your manuscript 'Maximizing multi-reaction dependencies provides more accurate and precise predictions of intracellular fluxes than the principle of parsimony' has been provisionally accepted for publication in PLOS Computational Biology.

Best regards,

Editorial Board

PLOS Computational Biology

Douglas Lauffenburger

%CORR_ED_EDITOR_ROLE%

PLOS Computational Biology

Reviewer's Responses to Questions

**Comments to the Authors:**

Reviewer #1: No further comments for Authors

Reviewer #2: The reviewer's comment has been addressed.

**Have the authors made all data and (if applicable) computational code underlying the findings in their manuscript fully available?**

Reviewer #1: None

Reviewer #2: Yes

PLOS authors have the option to publish the peer review history of their article (what does this mean?). If published, this will include your full peer review and any attached files.

Reviewer #1: No

Reviewer #2: **Yes: **Rajib Saha

---

## [Editor Report · Acceptance letter]

14 Sep 2023

PCOMPBIOL-D-22-01807R2 

Maximizing multi-reaction dependencies provides more accurate and precise predictions of intracellular fluxes than the principle of parsimony

Dear Dr Nikoloski,

I am pleased to inform you that your manuscript has been formally accepted for publication in PLOS Computational Biology. Your manuscript is now with our production department and you will be notified of the publication date in due course.

With kind regards,

Zsofia Freund
